# Uncooled sub-GHz spin bolometer driven by auto-oscillation

Minori Goto [1,2 ✉], Yuma Yamada [1], Atsushi Shimura[3], Tsuyoshi Suzuki[3], Naomichi Degawa[3], Takekazu Yamane[3], Susumu Aoki[3], Junichiro Urabe[3], Shinji Hara[3], Hikaru Nomura[1,2] & Yoshishige Suzuki[1,2]

Bolometers are rectification devices that convert electromagnetic waves into direct current voltage through a temperature change. A superconducting bolometer has a responsivity of approximately $10^6$–$10^7$ V/W under cryogenic temperatures at infrared wavelengths; however, no devices have realized such a high responsivity in the sub-GHz frequency region. We describe a spin bolometer with a responsivity of $(4.40 \pm 0.04) \times 10^6$ V/W in the sub-GHz region at room temperature using heat generated in magnetic tunnel junctions through auto-oscillation. We attribute the unexpectedly high responsivity to a heat-induced spin-torque. This spin-torque modulates and synchronizes the magnetization precession due to the spin-torque auto-oscillation and produces a large voltage output. In our device, heat-induced spin-torque was obtained because of a large heat-controlled magnetic anisotropy change: $-2.7$ μJ/Wm, which is significant for enhancing dynamic range and responsivity. This study can potentially lead to the development of highly sensitive microwave detectors in the sub-GHz region.

[1] Graduate School of Engineering Science, Osaka University, 1-3, Machikaneyamacho, Toyonaka, Osaka 560-8531, Japan. [2] Center for Spintronics Research Network (CSRN), Graduate School of Engineering Science, Osaka University, 1-3, Machikaneyamacho, Toyonaka, Osaka 560-8531, Japan. [3] TDK Corporation, 2-5-1 Nihonbashi, Chuo-ku, Tokyo 103-6128, Japan. ✉email: goto@mp.es.osaka-u.ac.jp

A bolometer is a rectification device that converts an electromagnetic wave into resistance or voltage change through heat generation;[1] such a device may be used to detect weak electromagnetic waves in radio astronomy and thermography[2]. Various types of bolometers such as semiconductor[3–5], superconductor[6–8], and carbon bolometers[9,10] have been developed for detecting infrared and millimeter wavelength radiation. Cooled bolometers using a graphene system have been developed over the last two decades and exhibit high responsivity:[11,12] Fatimy et al. have reported a very high responsivity of $5 \times 10^{10}$ V/W at 2.5 K[11]. In addition, a superconductor |insulator|normal metal|insulator|superconductor (SINIS) system bolometer was developed in 2000[13]. Recently, arrays of bolometers of these types at cryogenic temperatures lower than 400 mK have been reported to show very high responsivities of the order of $10^9$ V/W ($10^7$ V/W for a single device)[14,15]. Uncooled amorphous Si–Ge type semiconductor bolometers also possess a high responsivity: $4.2 \times 10^6$ V/W at room temperature[16,17]. Recently, a $VO_2$ coated carbon nanocoil[18] and a nano-mechanical bolometer using graphene[19] have shown responsivities of 3.3 and $5.9 \times 10^5$ V/W, respectively. However, conventional bolometers have not been utilized in the sub-GHz frequency region (Fig. 1a)[11,14–17,19].

In this frequency region, a similar rectification effect can be realized in diode devices such as the conventional Schottky barrier diode. A recent study reported that the responsivity of devices based on the spin-torque diode effect using magnetic tunnel junctions (MTJs) exceeds that of Schottky barrier diodes. The spin-torque diode effect was first reported in 2005; the responsivity was 0.5 V/W[20]. Following this, responsivities higher than those of Schottky barrier diodes were reported using various types of MTJs[21–25]. Zhang et al. reported responsivities of up to $2 \times 10^5$ V/W[25] using spin-torque auto-oscillation[26] in MTJs. However, a responsivity as high as $10^6$ V/W has never been realized in a rectification device operating in the sub-GHz frequency region as can be seen from Fig. 1a[21–25].

In this research, we developed a spin bolometer showing average and maximum responsivities of $(1.87 \pm 0.09) \times 10^6$ V/W and $(4.40 \pm 0.04) \times 10^6$ V/W, respectively, at room temperature using heat generation in MTJs. We showed that this high responsivity is attributable to the heat-controlled magnetic anisotropy (HCMA)[27]. It was not obvious in advance that heat-induced spin torque would provide a high responsivity exceeding that of spin-transfer torque[25], given the difference between the directions of their spin torques. Nevertheless, we have found this to be the case. It is remarkable to note that the mechanism of dc voltage generation through resistance change due to heat generation is the same as that observed in conventional bolometers. This paper describes a highly sensitive microwave detection technique using heat-induced spin torque.

Figure 1b, c shows the schematics of a circuit and an MTJ without and with the application of microwaves. MTJs have the structure: ferromagnet (pinned layer)|insulator|ferromagnet (free layer)|insulator. The magnetizations of the free and pinned layers are represented by red and black arrows. Bias voltage is applied to the MTJ, which induces the magnetization precession of the free layer due to spin-torque auto-oscillation[26]. As shown in Fig. 1c, the application of microwaves to the MTJ changes the temperature of the free layer. This induces a change in the magnetization precession, and, as a result, the MTJ's resistance is changed. With a dc bias current, this change in the resistance can be detected by the dc voltage change (see Supplementary Note 1 for details).

## Results

**Experimental design.** In this study, an MTJ with the structure "Bottom electrode|Buffer layer (Ru | Ta)|Ir–Mn (7.0 nm) | Co–Fe| Ru|Co–Fe–B|MgO (1.0 nm)|Fe–B (2.0 nm)|MgO (0.5 nm)|Metal cap (Ta (3.0 nm)|Ru (7.0 nm))|Top electrode" was deposited on a silicon substrate (see "Methods" section). The diameter of the MTJ was 190 nm. Figure 2a shows the schematic of the MTJ, along with a coordinate system with azimuthal angle $\theta$ and rotation angle $\varphi$. The red and black arrows represent free and pinned layer magnetization, respectively (the pinned-layer magnetization is in the $-y$ direction.) We obtained a magnetoresistance ratio of 43% and a resistance-area product of 3.9 $\Omega \, \mu m^2$ (see Supplementary Note 2 for further details). The circuit used for the spin-torque diode measurement is shown in Supplementary Note 2.

**Responsivity.** Figure 2b shows the frequency dependence of the diode voltage under a dc current of −2.6 mA and microwaves with a power of −55 dBm. A clear resonance peak was observed at 0.62 GHz. To obtain the maximum diode voltage, we measured its magnitude under various conditions of the applied magnetic field (Fig. 2c). We obtained a maximum diode voltage of 10.6 mV at an optimal condition of $B = 50$ mT, $\theta = 11°$, and $\varphi = 45°$ at $I_{dc} = −2.6$ mA. From this diode voltage, we obtained a responsivity (diode voltage/input microwave power) of $(3.37 \pm 0.03) \times 10^6$ V/W. Taking into account an insertion loss of the cables and bias-T of 1.16 dB, the responsivity was $(4.40 \pm 0.04) \times 10^6$ V/W. This value of responsivity is approximately twenty times higher than a previously reported value of the spin-torque diode effect[25].

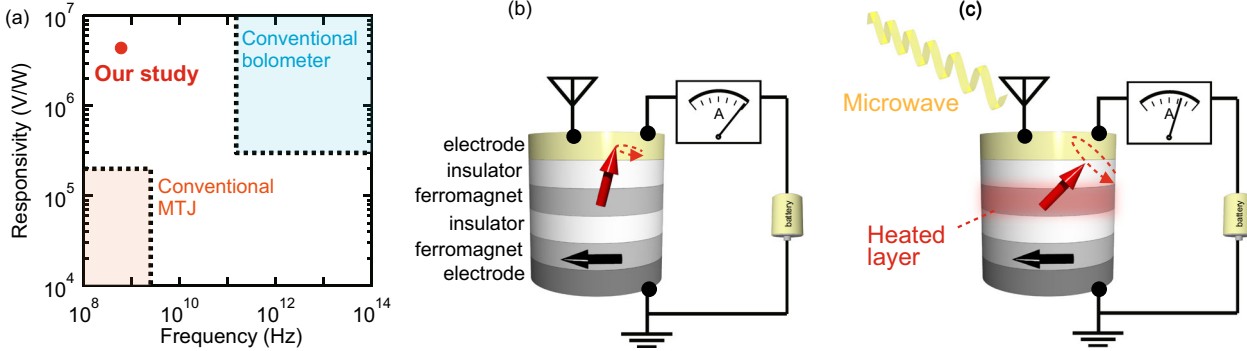

**Fig. 1 Positioning of our study in terms of responsivity with respect to the frequency map, and mechanism of the spin bolometer. a** Responsivities of rectification devices vs frequency. Blue and orange rectangular regions refer to bolometers and magnetic tunnel junctions (MTJs), respectively. Red point represents our result. **b**, **c** Schematic of spin bolometer (**b**) without and (**c**) with applied microwaves. Red and black arrows represent the magnetizations of the ferromagnetic free and pinned layers, respectively.

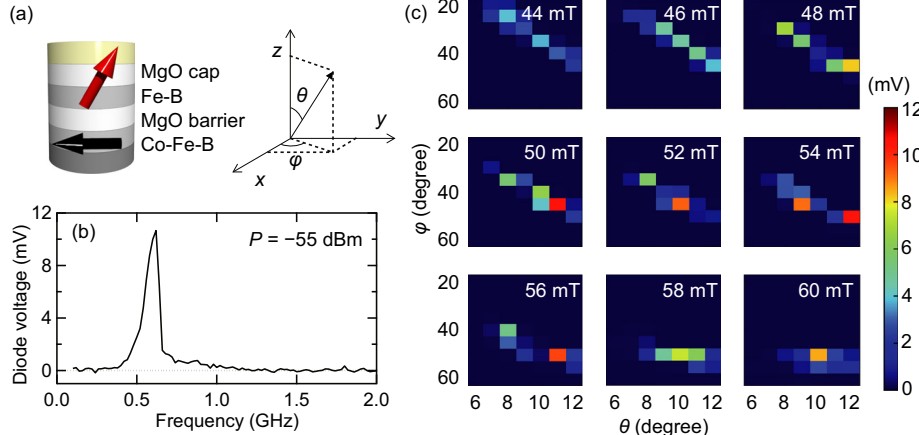

**Fig. 2 Spin-torque diode spectrum. a** Schematic of a magnetic tunnel junction and a coordinate system with azimuthal angle $\theta$ and rotation angle $\varphi$. The red and black arrows represent the free and pinned layer magnetizations, respectively. **b** Frequency dependence of diode voltage under magnetic field $B =$ 50 mT, $\theta = 11°$, and $\varphi = 45°$ at a dc bias current $I_{dc} = -2.6$ mA. Input microwave power $P$ is $-55$ dBm. **c** Color mapping of diode voltages under magnetic field intensities of 44–60 mT, azimuthal angles of 6°–12°, and rotation angles of 20°–60°.

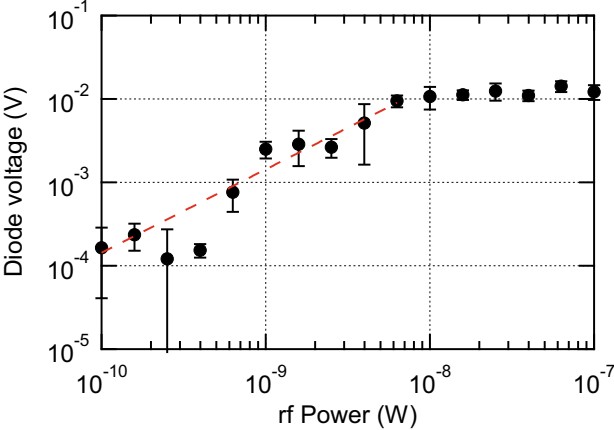

**Fig. 3 Microwave power dependence of diode voltage under the optimal condition (magnetic field intensity $B = 50$ mT, azimuthal angle $\theta = 11°$, and rotation angle $\varphi = 45°$).** The errors are defined as the standard deviations of the diode voltage in the vicinity of the resonance frequency, 0.60–0.62 GHz. The dashed red line represents the linear fitting of diode voltages less than 10 nW.

Figure 3 shows the microwave power dependence of the diode voltage under optimal conditions. The diode voltage monotonically increases with microwave powers less than 10 nW and saturates at microwave powers above 10 nW. The dashed red line represents the linear fitting of diode voltages less than 10 nW, which corresponds to a responsivity of $(1.87 \pm 0.09) \times 10^6$ V/W including the insertion loss of the circuit. This value is one order higher than that obtained earlier[25] and is similar to the responsivity obtained in conventional uncooled bolometers[16–19].

To understand the origin of this diode voltage, we measured its dependence on the elevation angle of the magnetic field. We found that this dependence is consistent with the symmetry of the nonlinear diode voltage (see Supplementary Note 3), suggesting that the high diode voltage is a result of the nonlinear diode effect.

**Noise power spectrum**. Further, we measured the noise power spectrum of MTJs under a dc bias current to confirm the excitation of spin-torque auto-oscillation. Figure 4a shows the circuit used for noise measurement (see "Methods" section). Noise

power from the MTJ was measured using a spectrum analyzer. Figure 4b shows the color mapping of the noise power spectrum under a dc bias voltage. We applied a magnetic field with $B = 48$ mT, $\theta = 19.3°$, and $\varphi = 80°$, resulting in an optimum diode voltage of 570 μV at $I_{dc} = -2.4$ mA and microwave power of $-56$ dBm. The dashed yellow line indicates magnoise, i.e., thermally excited magnetic noise; its frequency corresponds to the ferromagnetic resonance frequency. We found that the peak frequency discontinuously changed at a bias voltage of approximately $-290$ mV. Figure 4c shows the noise power spectra in the voltage range of $-280$ to $-299$ mV. The magnoise peak shown by the dashed yellow line disappears at $-295$ mV, and the new peak shown by the dashed red line appears at $-290$ mV. As discussed below, we attributed the new peak to spin-torque auto-oscillation.

We evaluated the bias-voltage dependence of the peak frequency, full width at half maximum, resistance, and diode voltage, as shown in Fig. 5a–d, respectively. The red dashed line represents the voltage $V_A$ at which the new peak appears, as shown in Fig. 5a. As shown in Fig. 5b, the full width at half maximum has its local minimum at a bias voltage of approximately $-180$ mV. This is typical behavior during spin-torque auto-oscillation and is evidence of an anti-damping torque. The dashed blue line shows a linear fitting in the range $-10$ to $-180$ mV. The horizontal intercept of this line corresponds to approximately $-260$ mV, which is the threshold of spin-torque auto-oscillation and is close to $V_A$. Moreover, the resistance of the MTJ drastically decreases at $V_A$. This is because the magnetization precession angle increases owing to spin-torque auto-oscillation, and the precession center is changed by a higher-order magnetization potential. This resistance change due to magnetization precession is the origin of the nonlinear diode voltage. As shown in Fig. 5d, $V_A$ corresponds to the threshold of the diode voltage. Therefore, the high diode responsivity is induced by spin-torque auto-oscillation.

**HCMA**. The nonlinear diode effect resulting from spin-torque auto-oscillation requires synchronization with a radio-frequency spin torque[28]. The possible origin of radio-frequency spin torque in this study is the HCMA. We have reported earlier that an extremely high spin-torque is generated by Joule heating in the MgO|FeB|MgO system[27]. To measure the magnitude of HCMA in our MTJ, we determined the magnetic anisotropy through spin-torque diode measurements. Figure 6a shows the

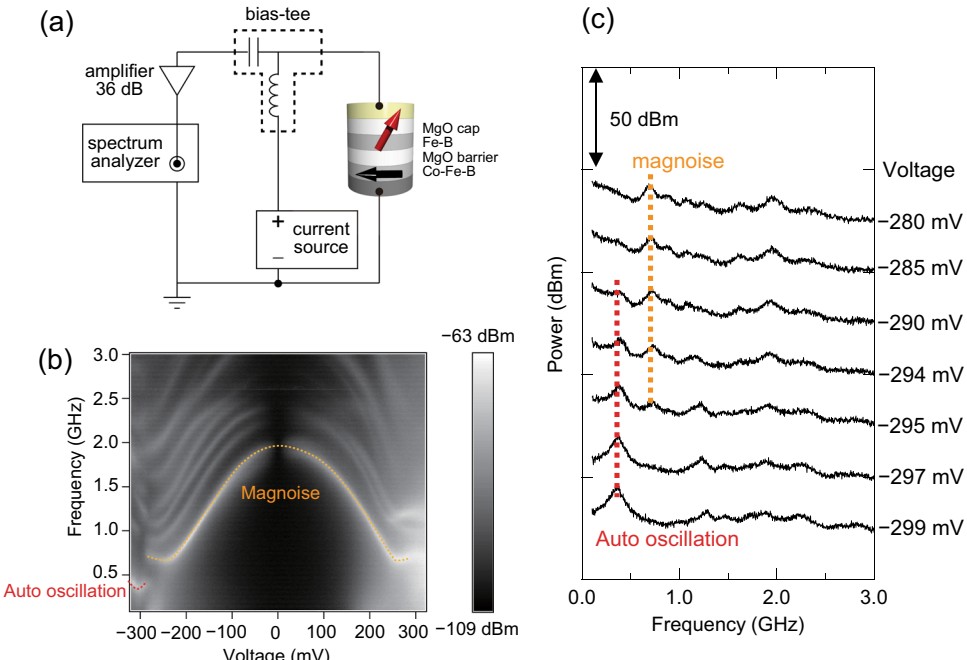

**Fig. 4 Noise power spectrum measurement. a** Schematic of measurement circuit used for noise power spectrum. **b** Color mapping of bias-voltage dependence of noise power spectrum. The resolution bandwidth is 5 MHz. **c** Noise power spectra in the bias-voltage range of −280 to −299 mV. In both (**b**) and (**c**), dashed yellow and red lines represent the peak frequency due to magnoise and spin-torque auto-oscillation, respectively. These are drawn in free hand as a guide for the eye.

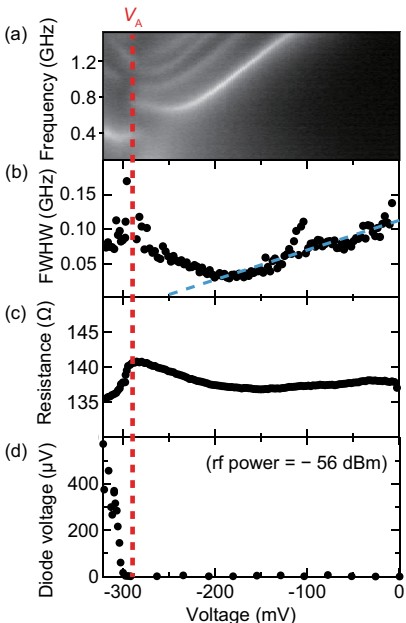

**Fig. 5 Influence of spin-torque auto-oscillation. a** Magnified graph of Fig. 4b. **b**–**d** Bias-voltage dependence of (**b**) full width at half maximum (FWHM), (**c**) resistance, and (**d**) diode voltage. The dashed red line is the threshold voltage $V_A$ associated with the diode effect. The dashed blue line is the linear fitting of the FWHM in the range of −180 to −10 mV; its horizontal intercept corresponds to the threshold voltage of spin-torque auto-oscillation.

perpendicular magnetic field dependence of the resonance frequency of magnetization. The measurement circuit is the same as that shown in Supplementary Fig. S1(a) (see "Methods" section and Supplementary Note 2). Microwaves were applied to the MTJ

with a power of −30 dBm at the sample position. The dashed lines indicate linear fitting of the resonance frequency, the horizontal intercept of which corresponds to the effective anisotropy field $\Delta H_{eff}$. We calculated the perpendicular magnetic anisotropy $K = \frac{1}{2}\mu_0 M_s \Delta H_{eff}$ using the effective anisotropy field and a saturation magnetization $M_s$ of 1.9 T. Figure 6b shows the bias-voltage dependence of the perpendicular magnetic anisotropy. The open and filled circles represent the voltage sweep direction. The red dashed line is the fitting curve of a second-order polynomial. We found that the perpendicular magnetic anisotropy parabolically decreases with the bias voltage. This result suggests that the perpendicular magnetic anisotropy is changed by Joule heating.

Using the second-order coefficient of the fitting equation, we obtained the magnitude of HCMA, equal to −2.7 μJ/Wm. HCMA is defined as the change in the perpendicular magnetic anisotropy by Joule heating per unit area:

$$\text{HCMA} = \frac{\partial K}{\partial (P/S)} = k_2 SR, \qquad (1)$$

where $P$ represents Joule heating, $S$ is the area of the MTJ, $R$ is the resistance of the MTJ, and $k_i$ ($i = 0, 1, 2$) is the polynomial coefficient of the bias-voltage dependence of perpendicular magnetic anisotropy, given by $K = K_0 + k_1 V + k_2 V^2$. In general, $k_2$ may include contributions from the nonlinear voltage-controlled magnetic anisotropy (VCMA) effect; for example, a V-shaped voltage dependence of magnetic anisotropy has been reported previously in an MgO|FeB|MgO-based high-RA MTJ[29]. However, as shown in Fig. 6(b), our low RA-MTJ shows a clear parabolic voltage dependence of magnetic anisotropy, and therefore, the nonlinear VCMA effect does not dominantly contribute to our result. This suggests that the contribution of Joule heating induced magnetic anisotropy change is dominant; accordingly, we calculated HCMA using Eq. (1).

The HCMA so obtained is approximately triple the previously reported value of −0.9 μJ/Wm[27]. The increase in HCMA value is

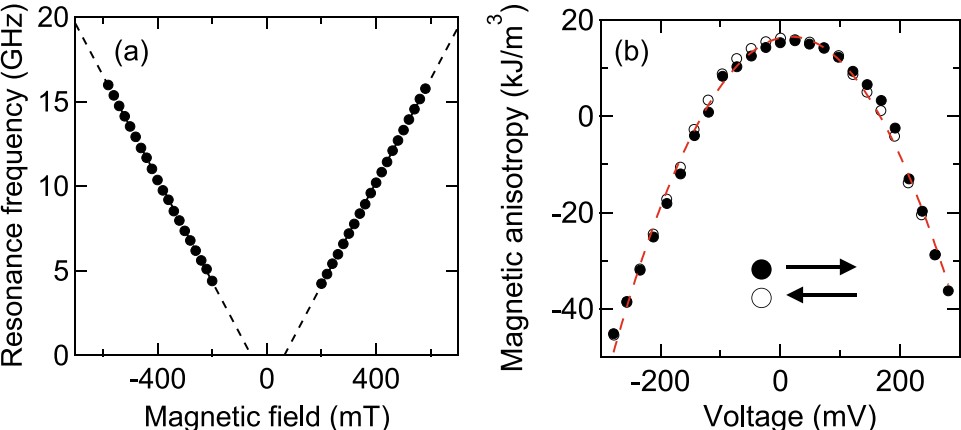

**Fig. 6 Characterization of HCMA from FMR measurement. a** Perpendicular magnetic field dependence of the ferromagnetic resonance frequency measured by the spin-torque diode technique. The dashed lines indicate linear fitting. **b** Bias-voltage dependence of perpendicular magnetic anisotropy. The open and filled circles represent the voltage sweep directions. The dashed red line is the fitting curve of a second-order polynomial.

attributable to the high-resistance MgO capping. Although the resistance of the MgO capping in the previous study was smaller than that of the MgO barrier, the two resistances are approximately the same in this study. The high-resistance MgO capping suppresses the diffusion of heat and enhances the temperature increase. As a result, the spin torque from the HCMA is larger than that from the VCMA effect and the spin-transfer torque (see Supplementary Note 4). We conclude that that the high diode voltage results from the heat-induced spin torque due to the high HCMA.

HCMA provides high responsivity in a certain range of rf power. In a previous study, a responsivity of $8 \times 10^4$ V/W was obtained using vortex core expulsion in a dc-biased MTJ[24]. However, when that technique is used, the diode voltage saturates with increasing microwave power, which suppresses the responsivity. By contrast, the device in the present study keeps its high responsivity across a relatively wide range of rf power, as shown in Fig. 3. Moreover, in the previous study, the responsivity of $2 \times 10^5$ V/W was obtained using spin-torque auto-oscillation and phase locking[25]. The present study uses HCMA rather than spin-transfer torque for phase locking. To compare each spin torque, we have characterized the magnitudes of spin torques in terms of their effective magnetic fields. In our devices, the effective rf magnetic fields of HCMA and spin-transfer torque are 500 μT and 9.8 μT, respectively, at the microwave power of −55 dBm (see Supplementary Note 4). This result suggests that the HCMA generates a larger spin torque than conventional spin-transfer torque, and thus provides higher responsivity.

Moreover, the HCMA value can be enhanced by the improvement of thermal design as discussed by Okuno[30]. By contrast, enhancement of spin-transfer torque requires a decrease in the magnetization or thickness of the ferromagnetic layer; this induces deterioration in MTJs. Therefore, utilization of HCMA is promising for further enhancement of responsivity.

HCMA is also useful for enhancement of dynamic range. Although the dynamic range is limited by the noise equivalent voltage, it can be improved by increasing the ferromagnetic thickness. However, spin-transfer torque and VCMA decrease significantly when this is done. HCMA decreases only slightly because the increase in the temperature of the ferromagnetic layer is mainly affected by the MgO layer through which the heat flows, rather than the FeB layer. Therefore, HCMA is the appropriate spin torque for improving dynamic range (see Supplementary Note 5).

**Noise equivalent power**. Finally, we discuss the noise equivalent power (NEP). We obtained a minimum NEP of $2.4 \times 10^{-12}$ W/$\sqrt{\text{Hz}}$ under optimal conditions of $B = 54$ mT, $\theta = 12°$, and $\varphi = 50°$ at a dc bias current of −2.6 mA and an input microwave power of −55 dBm including the insertion loss of 1.16 dB (see Supplementary Note 6). This is five to six orders higher than the previously obtained NEP of $10^{-17} \sim 10^{-18}$ W/$\sqrt{\text{Hz}}$ using 192 cold electron bolometer arrays at a cryogenic temperature of approximately 260 mK[14]. We can obtain an NEP of $1.5 \times 10^{-16}$ W/$\sqrt{\text{Hz}}$ using the same number of devices and under the same temperature. Then, our NEP becomes only one to two orders larger than that of a superconducting bolometer, although the detecting frequency is two to three orders smaller. Hence, to improve the NEP of the spin bolometer, suppression of magnetic noise and enhancement of responsivity are needed. While the NEP under the nonlinear diode effect in an MTJ has been discussed in a previous study[23], the NEP under the spin-torque auto-oscillation has not. Therefore, further theoretical investigation is necessary in this regard.

We have developed a spin bolometer using heat generation in MTJs. We obtained average and maximum responsivities of $(1.87 \pm 0.09) \times 10^6$ V/W and $(4.40 \pm 0.04) \times 10^6$ V/W, respectively. This device produces a high responsivity value that has never been reported before at room temperature in the sub-GHz frequency range. This high responsivity is attributed to the nonlinear diode voltage generated by heat-induced spin torque under spin-torque auto-oscillation conditions. The spin-torque auto-oscillation is synchronized with the heat-induced spin torque because of the high HCMA. In this study, we have shown that the HCMA is useful for obtaining high responsivity. Moreover, we obtained an HCMA value of −2.7 μJ/Wm, which is approximately triple that obtained previously. Using HCMA, further improvements in dynamic range and responsivity are possible. Our study also shows that the heat-induced spin torque is significant for microwave characteristics of spin devices. In the future, further research is required to improve the NEP under spin-torque auto-oscillation conditions. This research will serve as the basis for developing highly sensitive microwave communication devices.

## Methods

**Sample preparation**. An MTJ (procured from TDK Corporation) with the structure "Bottom electrode | Buffer layer (Ru | Ta)|Ir–Mn (7.0 nm)|Co–Fe|Ru| Co–Fe–B|MgO (1.0 nm)|Fe–B (2.0 nm)|MgO (0.5 nm)|Metal cap (Ta (3.0 nm)|Ru (7.0 nm))|Top electrode" was deposited via magnetron sputtering on a silicon substrate measuring 20 × 20 × 0.5 mm. An MTJ with a design diameter of 190 nm

was fabricated via photolithography. Pinned-layer magnetization occurred along the $-y$ direction.

**Diode measurement**. For the diode measurements, microwaves were applied to the MTJ from a signal generator (Keysight, E8257D). The amplitude modulation frequencies of microwaves were 7.5 kHz, 1.0 kHz, and 3.0 kHz in Figs. 2–3, 5d, 6, respectively. The diode voltage was measured by a lock-in amplifier (Stanford Research Systems, SR830) synchronized with the signal generator. For the measurements shown in Figs. 2, 3, 5d, an attenuator (Keysight 8493 C) of −50 dB was used to reduce the intensity of the microwaves. For the measurements shown in Fig. 6, an attenuator of −30 dB was used. In Fig. 3, the errors are defined as the standard deviations of diode voltages in the vicinity of the resonance frequency, 0.60–0.62 GHz. The errors of average and maximum responsivities are defined as the deviation of linear fitting in Fig. 3 and the standard deviation of diode voltage in the range 1.3 GHz to 2.0 GHz, respectively. For characterizing perpendicular magnetic anisotropy, the saturation magnetization of 1.9 T was measured using a vibrating sample magnetometer. We used a SUCOFLEX 104 coaxial cable with a frequency span of up to 26.5 GHz and the bias-T of the Keysight 11612B bias network with a frequency span of 0.045–50 GHz. The total insertion loss was 1.16 dB at 0.6 GHz.

**Measurement of noise power spectrum**. Noise power was amplified by an amplifier with a gain of 36 dB and was measured using a spectrum analyzer (Keysight E4448A) with a resolution bandwidth of 5 MHz. We applied a magnetic field with $B = 48$ mT, $\theta = 19.3°$, and $\varphi = 80°$, which yielded the optimum diode voltage of 570 μV at $I_{dc} = -2.4$ mA and microwave power −56 dBm.

**Correspondence of samples and measurement results**. The results shown in Figs. 4, 5 were obtained by measuring a different MTJ device having the same design as the one used to record data in Figs. 2, 3. To clarify further: one sample was used to measure the frequency and microwave power dependencies of the diode voltage (Figs. 2, 3); the other was used for the noise power spectrum and diode measurements (Figs. 4, 5, 6).

## Data availability

The data that support the plots within this paper and other findings of this study are available from the corresponding author upon reasonable request.

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

## Acknowledgements

This work was supported by JSPS KAKENHI Grant Number JP19K15435. M.G., H.N., and Y.S. are members of Center for Spintronics Research Network (CSRN), the Osaka University, under Spintronics Research Network of Japan (Spin-RNJ). We would like to thank Editage for English language editing.

## Author contributions

M.G. and Y.Y. conducted the measurements. A.S., T.S., N.D., T.Y., S.A., J.U., and S.H. prepared the sample. M.G. and H.N. conducted the analysis. Y.S. developed the explanation of the experimental result. All authors discussed the results and commented on the manuscript.

## Competing interests

The authors declare no competing interests.
