## [Peer Review File · Nature Communications]

REVIEWER COMMENTS

Reviewer #1 (Remarks to the Author):

Minori Goto et al have reported the spin bolometer with a responsivity of 10^6 V/W in the sub-GHz region at room temperature using heat generation in MTJs through auto-oscillation. It is a new microwave detection technique to obtain such high responsivity at microwave power less than 10 nW. While I find the work is an interesting advancement in microwave detection, the authors need to address some issues before the manuscript can be further considered in Nature Communications.

1. As the authors mention in the abstract, a bolometer is a rectification device utilized in infrared detector. The infrared detector has high detection frequency (about 60 GHz-30 THz) and measures the change in temperature. In this study, the mechanism of dc voltage generation is related to resistance change due to heat generation, but the high responsivity occurs at sub-GHz (about 0.6 GHz). Hence, I wonder whether it is possible to achieve high responsivity in the high frequency region (for instance, 10 GHz) by nonlinear diode effect?

2.The authors attribute this high responsivity to the heat-induced spin-torque, and the heat-induced spin-torque is obtained because of a large heat-controlled magnetic anisotropy (HCMA) change. It would be better if the authors can address more about the relationship between responsivity and HCMA.

3.In previous studies, the high responsivity can be obtained by the heat-induced spin-torque, spin-torque resonant expulsion of the vortex core and phase-locking technique. Please the authors address the differences among them, or how to distinguish the role of HCMA from others since all cases involved the application of DC bias.

4.The obtained HCMA in this study is more than double their previous report [27]. Why is the HCMA enhanced? Is it related to the thickness of FeB or MgO ?

5. How do the authors consider the power loss during frequency span in microwave circuits?

Reviewer #2 (Remarks to the Author):

This paper presents interesting results on the use of an auto-oscillating magnetic tunnel junction for detection of rf electromagnetic fields through bolometric means. The results are illustrated through point-by-point illustration of various properties of the device, backed up by data plots and interpretation that are fairly compelling. The authors attribute the observed order of magnitude increase in the responsivity over previous results to a “heat controlled magnetic anisotropy” (HCMA), which apparently has not been utilized for rf detection before. However, my overall impression is that this work is an incremental improvement over previous work (Ref. 25) and therefore is neither surprising nor a breakthrough worthy of a high profile journal such as Nature Communications. The results in Ref. 25 from 2018 are nearly as impressive as those described here,

and also do not require an external bias magnetic field. The authors do not make a compelling case that the present work is significantly different or qualitatively improved over that previously reported. At the face of it, there is merely a quantitative improvement of responsivity from 2×10^5 V/W to 3×10^6 V/W. Is there a promise now for dramatically improved performance? Has some sort of revolutionary new effect been discovered, or is this just an improvement of existing technology? After reading this paper several times I have to answer these questions in the negative. However, one important contribution of this work is the further elucidation of the detailed mechanism of the bolometric effect. The utilization of HCMA as an important ingredient in the improved responsivity is noteworthy. However, this mechanism has already been identified and published by the authors in Ref. [27], where they extensively analyzed a very similar device in a very similar set of measurements. Again the present manuscript seems to be simply an incremental improvement over previously published work.

The paper reads as a rather dry technical report rather than a Nature Communications article. The authors discuss the mechanism for the observed effects on pages 4 and 5, but many readers will find this simply to be a list of jargon that only experts in this field can appreciate or understand. Instead, I recommend that the authors introduce a new figure in the form of a creatively illustrated drawing that clearly illustrates the mechanism for the effect through un-ambiguous near-cartoonish depiction. The reader should be drawn in by this illustration and left with a clear understanding of the basic physics that leads to the observed effect. This will also give the authors the opportunity to clearly state what is unique and original about this device design and operation, and what revolutionary new breakthrough has been made.

I have a few more detailed concerns about the impact of these results, especially those shown in Figs. 1 and 2. First, the voltage response is strong only over a very narrow frequency range. How will the detector be made to work over a broad frequency band? Second, the voltage response is linear only over a relatively narrow dynamic range of microwave signal strength. How will the dynamic range of the detector be improved?

REVIEWER COMMENTS

Reviewer #1 (Remarks to the Author):

Minori Goto et al have reported the spin bolometer with a responsivity of 10^6 V/W in the sub-GHz region at room temperature using heat generation in MTJs through auto-oscillation. It is a new microwave detection technique to obtain such high responsivity at microwave power less than 10 nW. While I find the work is an interesting advancement in microwave detection, the authors need to address some issues before the manuscript can be further considered in Nature Communications.

1. As the authors mention in the abstract, a bolometer is a rectification device utilized in infrared detector. The infrared detector has high detection frequency (about 60 GHz-30 THz) and measures the change in temperature. In this study, the mechanism of dc voltage generation is related to resistance change due to heat generation, but the high responsivity occurs at sub-GHz (about 0.6 GHz). Hence, I wonder whether it is possible to achieve high responsivity in the high frequency region (for instance, 10 GHz) by nonlinear diode effect?

Response 1-1

Thank you for your comment. If we change the magnetic field and voltage conditions, the peak frequency can be tuned as shown in Fig R1. Therefore, in principle, the nonlinear diode voltage can be measured at 10 GHz. However, generally, the responsivity decreases with increasing frequency. Therefore, it is difficult to maintain high responsivity at 10 GHz. Our device is the bolometer that shows high responsivity in the sub-GHz frequency range. To avoid misleading readers, we change the title as follows.

Figure R1 Frequency dependence of diode voltage under various magnetic field conditions (B , θ , φ). Red, green, and blue curves represent the diode spectra at the magnetic field conditions (50 mT, 79°, 45°), (54 mT, 78°, 50°), and (60 mT, 80°, 50°), respectively.

Before correction: Title

Uncooled Radio-frequency Spin Bolometer Driven by Auto-oscillation

After correction: Title

Uncooled sub-GHz Spin Bolometer Driven by Auto-oscillation

Before correction: Main text, line 48

In the radio-frequency region,

After correction: Main text, line 49

In this frequency region,

Before correction: Main text, line 56

in the radio-frequency region

After correction: Main text, line 57

in the sub-GHz frequency region as can be seen from Fig 1(a)²¹⁻²⁵.

2.The authors attribute this high responsivity to the heat-induced spin-torque, and the heat-induced spin-torque is obtained because of a large heat-controlled magnetic anisotropy (HCMA) change. It would be better if the authors can address more about the relationship between responsivity and HCMA.

Response 1-2

Thank you for making this constructive comment on the need to discuss the relation between HCMA and responsivity. We have considered the relation between responsivity and HCMA. From the power dependence of responsivity as shown in Fig 3, the diode voltage is proportional to the microwave power. Moreover, the heat induced spin-torque in dc-biased MTJ is proportional to the microwave voltage, implying that the diode voltage is proportional to the square of rf spin-torque. Therefore, responsivity is proportional to the square of the HCMA. We have mentioned this in Supplementary Information.

In addition, we have attempted to experimentally verify the relation between responsivity and HCMA. As reported by Okuno, HCMA can be controlled by the thickness of MgO capping [R. Okuno *et al.*, J. Phys. Cond. Mater (2020)]. In the present study, we characterize the responsivity of the MTJ with the MgO capping thickness of 0.3 nm, and compare it with the MTJ of 0.5 nm thick MgO capping. However, we have concluded that it is difficult to explain the relation between responsivity and HCMA in a simple way because the optimal conditions change. Therefore, we include this explanation only in the response letter:

Figure R2(a) shows the in-plane magnetic field dependence of the MTJ resistance. We characterize the bias-voltage dependence of the perpendicular magnetic anisotropy through spin-torque FMR measurement as shown in Fig R2(b). Curves with white and black circles represent measurements in the two voltage-sweep directions. The result shows the parabolic voltage dependence of

magnetic anisotropy. The red dashed line represents the second-order polynomial fitting curve.

Using the second-order coefficient k_2 in Eq (1), we obtained an HCMA of $-2.1 \mu\text{J/Wm}$. Using this MTJ, we found that the magnetic anisotropy has a hysteresis depending on the voltage sweep direction. This result suggests that ions in the MgO barrier or capping are migrating in response to the voltage. This ion migration does not often exert spin-torque because of its slow response. In this study, the HCMA is dominant since the magnetic-anisotropy change due to Joule heating is sufficiently large compared to that due to voltage hysteresis. We measured the diode voltage under various magnetic-field and bias-voltage conditions and found that the optimal condition is $B = -112 \text{ mT}$, $\theta = 82^\circ$, $\varphi = 286^\circ$, $V = -373 \text{ mV}$.

Figure R2(c) shows the microwave power the dependence of diode voltage under the optimal condition. When the power is less than 10^{-9} W , the diode voltage is smaller than the noise; above this threshold, it increases as a function the microwave power. The red dashed line represents the result of fitting with a function o the form Cp^λ ; we obtained $\lambda=0.74$. The highest responsivity ($1.74 \times 10^5 \text{ V/W}$) is obtained at 10^{-9} W except in the region that noise is dominant. By taking into consideration the insertion loss of 1.16 dB , we obtained a responsivity of $2.27 \times 10^5 \text{ V/W}$. This value is approximately $1/20$ of $4.40 \times 10^6 \text{ V/W}$, the responsivity in the MTJ with an HCMA of $-2.7 \mu\text{J/Wm}$. (The HCMA values have changed because we have corrected a mistake on the horizontal axis in Fig 5(b) of previous manuscript. This correction is mentioned in the last section of this response letter.) This difference cannot be explained by the HCMA change alone, because the diode voltage is proportional to at most the square of the spin-torque even in the case of $\lambda = 1$. The cause of the difference may be a change in the magnetic anisotropy and withstand voltage caused by the change in the capping layer thickness; this could affect the optimal conditions of magnetic field and bias voltage. Thus, it is clearly difficult to find the exact relation between the HCMA and the responsivity in general. However, if we limit the discussion to this experiment, we can say that we found responsivity to be high in an MTJ with high HCMA.

Figure R2. (a) The in-plane magnetic field dependence of MTJ resistance. Solid and dotted lines represent the direction of magnetic field sweep. (b) Bias-voltage dependence of perpendicular magnetic anisotropy. Open and filled circles represent the voltage sweep directions. The red dashed line is the second-order polynomial fitting result. (c) Microwave power dependence of the diode voltage. The red dashed line represents the result of fitting.

Correction 1-2-1

Insertion: Supplementary Information, line 145

Although we suggested that large HCMA (large rf-spin-torque) produces large responsivity, it is not obvious under the mechanism in this manuscript. Thus, here, we discuss the relation between rf-spin-torque and responsivity induced by HCMA. From the power dependence of responsivity as shown in Fig 3, larger spin-torques induce higher responsivity. The diode voltage is proportional to the microwave power, and heat induced spin-torque in dc-biased MTJ is proportional to the microwave voltage, implying that the diode voltage is proportional to the square of rf spin-torque. In our system, because the HCMA is high and exerts a large spin-torque, the diode voltage increases, and a high responsivity is obtained.

3.In previous studies, the high responsivity can be obtained by the

heat-induced spin-torque, spin-torque resonant expulsion of the vortex core and phase-locking technique. Please the authors address the differences among them, or how to distinguish the role of HCMA from others since all cases involved the application of DC bias.

Response 1-3

Thank you for your comment. As you mentioned, high responsivity has been obtained by spin-torque resonant expulsion of the vortex core and the phase-locking technique in previous studies. However, high responsivity obtained by heat-induced spin-torque is only now being reported, in this manuscript.

As you mentioned, each technique (vortex expulsion, phase-locking, and HCMA) needs bias-voltage. We explain the difference among them.

In one previous study (Zhang, L. *et al.*, Appl. Phys. Lett, **113**, 102401 (2018)), a high responsivity was obtained by spin-torque auto-oscillation and its phase-locking, which is same technique as ours. However, the rf spin-torque utilized for the phase-locking was different. The previous study used spin-transfer torque; we use HCMA, which generates a larger spin-torque. The large spin-torque produces the high responsivity in our study. To confirm the magnitude of the spin-torque, we observed the effective fields of each spin-torque by diode measurement. The effective fields of rf spin-transfer torque and rf HCMA are 9.8 μT and 500 μT , respectively, at the microwave power of -55 dBm. The large spin-torque induced by HCMA produces high responsivity.

Moreover, high responsivity has also been reported using the vortex-expulsion technique (S. Tsunegi *et al.*, Appl. Phys. Express, **11**, 053001 (2018)), in which the vortex core is expelled from the disk and disappears. After the vortex core is gone, the diode voltage saturates even if the microwave power increases. In the previous study, 80,000 V/W was shown at the power of 0.3 μW , after which the diode voltage saturates and does not increase further. On the other hand, with our MTJ, the diode voltage increases with increasing microwave power and keeps its high responsivity over a certain range of microwave power. This is the advantage of our bolometer.

To clarify these points, we have added the following explanations:

Correction 1-3-1

Insertion: Main text, line 176

HCMA provides high responsivity in a certain range of rf power. In a previous study, a responsivity of 8×10^4 V/W was obtained using vortex core expulsion in a dc-biased MTJ²⁴. However, when that technique is used, the diode voltage saturates with increasing microwave power, which suppresses the responsivity. By contrast, the device in the present study keeps its high responsivity across a relatively wide range of rf power, as shown in Fig 3. Moreover, in the previous study, the responsivity of 2×10^5 V/W was obtained using spin-torque auto-oscillation and phase locking²⁵. The present study uses HCMA rather than spin-transfer torque for phase locking. To compare each spin-torque, we have characterized the magnitudes of spin-torques in terms of their effective magnetic fields. In our devices, the effective rf magnetic fields of HCMA and spin transfer torque are 500 μ T and 9.8 μ T, respectively, at the microwave power of -55 dBm (see Supplementary Information). This result suggests that the HCMA generates a larger spin-torque than conventional spin-transfer torque, and thus provides higher responsivity.

Correction 1-3-2

Insertion: Supplementary Information, line 109

Effect of spin-torques on responsivity

We discuss the effect of various spin-torques, such as HCMA, VCMA, and spin-transfer torque, on the responsivity. Using the HCMA value calculated from Fig 6(b), the oscillation amplitude of the magnetic anisotropy field

$$\frac{2\Delta K_{HCMA}}{M_s} = \frac{4V_{dc}V_{ac}}{M_s SR} HCMA$$
 can be evaluated as 500 μ T at a microwave power of

-55 dBm with an insertion loss of 1.16 dB. Here, ΔK_{HCMA} is the oscillation

amplitude of magnetic anisotropy; M_s is the saturation magnetization; V_{dc} and

V_{ac} are the dc and ac voltage, respectively; and S and R are the area and resistance of the MTJ, respectively. In the same manner, the linear VCMA value can be calculated using k_1 , the coefficient of the linear term in the expansion of the perpendicular magnetic anisotropy (Fig 6(b)). The oscillation amplitude of the magnetic anisotropy field $\frac{2\Delta K_{VCMA}}{M_s} = \frac{2V_{ac}}{M_s t_{FeB} t_{MgO}} VCMA$ can be calculated to be 21.5 μ T under the same conditions.

Spin-transfer torque can be characterized by the amplitude of the diode voltage under a perpendicular magnetic field because the anisotropy change does not affect the magnetization dynamics in this condition. Figure S4(a) shows the diode spectra of an MTJ of diameter of 130 nm under various perpendicular magnetic fields at the microwave power of -25 dBm without bias voltage. These results are fitted by the function

$$V_{diode} = A \frac{\Delta f \cdot f^2}{(f^2 - f_r^2)^2 + (\Delta f \cdot f)} \quad (S1)$$

Here, f , f_r , and Δf represent the frequency of the microwave, the resonant frequency, and the full width at half maximum of the resonant peaks, respectively; A is the amplitude of the peak. Figure S4(b) shows the magnetic field dependence of the amplitude A . By solving Landau-Lifshitz-Gilbert-Slonczewski equation, the amplitude is calculated to be

$$A = \frac{g I_{ac}}{4\pi G_0} \gamma B_{stt}, \text{ where } I_{ac} \text{ is ac current, } \gamma \text{ is gyro-magnetic ratio, and } B_{stt} \text{ is}$$

the effective field of spin-transfer torque. The quantities g and G_0 are

$$\frac{G_p - G_{AP}}{G_p + G_{AP}} \text{ and } \frac{G_p + G_{AP}}{2}, \text{ respectively, where } G_p \text{ and } G_{AP} \text{ are the parallel and}$$

anti-parallel conductances, respectively. We used the amplitude at 100 mT because, as shown in Fig S1(c), the free-layer magnetization is almost saturated at that field intensity, and a larger magnetic field increases the modulation of the

pinned-layer magnetization. Using the amplitude of $3.4 \mu\text{VGHz}$ at the 100 mT, we obtained the effective magnetic field of spin-transfer torque $B_{\text{stt}} = 308 \mu\text{T}$ at a microwave power of -25 dBm , including an insertion loss of 1.16 dB . This value can be converted to $B_{\text{stt}} = 9.8 \mu\text{T}$ at a microwave power of -55 dBm , the condition of the experiment shown in Fig 3. Therefore, the spin-torque due to HCMA is dominant in this experiment.

Supplementary Figure S4. (a) Frequency dependence of diode voltage under various perpendicular magnetic fields. (b) Perpendicular magnetic field dependence of peak intensity A obtained by the fitting.

Reference

- 1 Miwa, S. *et al.* Highly sensitive nanoscale spin-torque diode. *Nat Mater* **13**, 50-56, doi:10.1038/nmat3778 (2014).
- 2 Kiselev, S. I. *et al.* Microwave oscillations of a nanomagnet driven by a spin-polarized current. *Nature* **425**, 380-383 (2003).
- 3 Goto, M. *et al.* Microwave amplification in a magnetic tunnel junction induced by heat-to-spin conversion at the nanoscale. *Nat Nanotechnol* **14**, 40-43, doi:10.1038/s41565-018-0306-9 (2019).
- 4 Zhang, L. *et al.* Ultrahigh detection sensitivity exceeding 105 V/W in spin-torque diode. *Applied Physics Letters* **113**, doi:10.1063/1.5047547 (2018).

4.The obtained HCMA in this study is more than double their previous report [27]. Why is the HCMA enhanced? Is it related to the thickness of FeB or

MgO ?

Response 1-4

Yes, the enhancement of HCMA is related to the thickness of the MgO capping. More accurately, the difference between the HCMA in ref [27] and in this study is caused by the resistance of the MgO capping. Although the previous paper did not mention it, this resistance was smaller than that of the MgO barrier. On the other hand, in the present study, the MgO capping had a high resistance comparable with the MgO barrier resistance. In general, since a high resistance film has a high thermal resistance because of the Wiedemann-Frantz law, our MTJ shows a higher HCMA value.

To clarify these points, we have added the following explanation in the main text:

Correction 1-4-1

Insertion: Main text, line 168

The HCMA so obtained is approximately triple the previously reported value of $-0.9 \mu\text{J}/\text{Wm}^{27}$. The increase in HCMA value is attributable to the high-resistance MgO capping. Although the resistance of the MgO capping in the previous study was smaller than that of the MgO barrier, the two resistances are approximately the same in this study. The high resistance MgO capping suppresses the diffusion of heat and enhances the temperature increase. As a result, the spin torque from the HCMA is larger than that from the VCMA effect and the spin-transfer torque (see Supplementary Information). We conclude that that the high diode voltage results from the heat-induced spin-torque due to the high HCMA.

5. How do the authors consider the power loss during frequency span in microwave circuits?

Response 1-5

Thank you for your valuable comment. We had not considered it. We used SMA cables (SUCOFLEX 104) with a frequency span of up to 26.5 GHz and a bias-T

(Keysight 11612b) with a frequency span of 0.045 GHz to 50 GHz. At the 0.6 GHz, the total insertion loss is 1.16 dB, which modulates the input power into 76.5 %. We have corrected the manuscript as follows:

Correction 1-5-1

Before correction: Main text and Supplementary Information, maximum responsivity

$$(3.37 \pm 0.03) \times 10^6 \text{ V/W}$$

After correction: Main text and Supplementary Information, maximum responsivity

$$(4.40 \pm 0.04) \times 10^6 \text{ V/W}$$

Correction 1-5-2

Before correction: Main text and Supplementary Information, average responsivity

$$(1.43 \pm 0.07) \times 10^6 \text{ V/W}$$

After correction: Main text and Supplementary Information, average responsivity

$$(1.87 \pm 0.09) \times 10^6 \text{ V/W}$$

Correction 1-5-3

Before correction: Main text and Supplementary Information, NEP with 192 devices under 260 mK

$$1.9 \times 10^{-16} \text{ W}/\sqrt{\text{Hz}}$$

After correction: Main text and Supplementary Information, NEP with 192 devices under 260 mK

$$1.5 \times 10^{-16} \text{ W}/\sqrt{\text{Hz}}$$

Correction 1-5-4

Insertion: Main text, line 95

Taking into account an insertion loss of the cables and bias-T of 1.16 dB, the responsivity was $(4.40 \pm 0.04) \times 10^6$ V/W.

Correction 1-5-5

Insertion: Main text, line 250

We used a SUCOFLEX 104 coaxial cable with a frequency span of up to 26.5 GHz and the bias-T of the Keysight 11612B bias network with a frequency span of 0.045 GHz to 50 GHz. The total insertion loss was 1.16 dB at 0.6 GHz.

Correction 1-5-6

Before correction: Supplementary Information, minimum NEP

a minimum NEP of $3.1 \times 10^{-12} \text{ W}/\sqrt{\text{Hz}}$

After correction: Supplementary Information, minimum NEP

a minimum NEP of $2.4 \times 10^{-12} \text{ W}/\sqrt{\text{Hz}}$

Correction 1-5-7

Before correction: Supplementary Information, line 97

an input microwave power of -55 dBm.

After correction: Supplementary Information, line 97

an input microwave power of -55 dBm including the insertion loss of 1.16 dB.

Correction 1-5-8

Insertion: Supplementary Information, line 102

Taking into consideration the insertion loss of 1.16 dB, the responsivity is $(4.35 \pm 0.01) \times 10^6$ V/W.

Correction 1-5-9

Before correction: Main text, line 178

at a dc bias current of -2.6 mA (see Supplementary Information)

After correction: Main text, line 202

at a dc bias current of -2.6 mA and an input microwave power of -55 dBm including the insertion loss of 1.16 dB (see Supplementary Information)

Reviewer #2 (Remarks to the Author):

This paper presents interesting results on the use of an auto-oscillating magnetic tunnel junction for detection of rf electromagnetic fields through bolometric means. The results are illustrated through point-by-point illustration of various properties of the device, backed up by data plots and interpretation that are fairly compelling. The authors attribute the observed order of magnitude increase in the responsivity over previous results to a “heat controlled magnetic anisotropy” (HCMA), which apparently has not been utilized for rf detection before. However, my overall impression is that this work is an incremental improvement over previous work (Ref. 25) and therefore is neither surprising nor a breakthrough worthy of a high profile journal such as Nature Communications. The results in Ref. 25 from 2018 are nearly as impressive as those described here, and also do not require an external bias magnetic field. The authors do not make a compelling case that the present work is significantly different or qualitatively improved over that previously reported. At the face of it, there is merely a quantitative improvement of responsivity from 2×10^5 V/W to 3×10^6 V/W. Is there a promise now for dramatically improved performance? Has some sort of revolutionary new effect been discovered, or is this just an improvement of existing technology? After reading this paper several times I have to answer these questions in the negative.

Response 2-1

Thank you for your comment. The main difference between this and previous studies is that we have reached a level of responsivity previously attained only by conventional bolometers, using a room-temperature MTJ device that operates at frequencies well below those of conventional bolometers. Moreover, the utilization of HCMA enables further improvement of the dynamic range and responsivity of the device in the future. The former is discussed here and the latter in Responses 2-2 and 2-4.

As you mentioned, the same technique of spin-torque auto-oscillation was used to obtain a large diode signal in Ref(L. Zhang *et al*, Appl. Phys. Lett, (2018)). However, going from 2.0×10^5 V/W to 4.40×10^6 V/W, as we do in this paper, is

not a mere incremental advance: it is the first time that bolometer-level responsivity has been achieved in the sub-GHz frequency region. (Note that the exact value of the responsivity has been corrected from 3×10^6 V/W to 4.40×10^6 V/W by considering the insertion loss of the circuit. This correction is mentioned in the response 1-5.) We show the regions of responsivity–frequency space explored by conventional bolometers and MTJs in Fig 1(a). The reported responsivity of conventional MTJs goes up to 2.0×10^5 V/W, which is close to the bottom of the responsivity range for conventional bolometers. On the other hand, conventional bolometers, despite their superior responsivity, operate only at infrared and millimeter wave frequencies. Our study achieves a bolometer-level responsivity of 4.40×10^6 V/W at room temperature in the sub-GHz frequency region. We believe that this is a revolutionary result paving the way to new applications.

Correction 2-1-1

Insertion: Main text, Figure 1(a)

Figure 1. (a) Responsivities of rectification devices vs frequency. Blue, yellow, and orange rectangular regions refer to bolometers and magnetic tunnel junctions (MTJs), respectively. Red point represents our result.

Correction 2-1-2

Insertion: Main text, line 47

However, conventional bolometers have not been utilized in the sub-GHz frequency region (Fig 1(a))^{11,14-17,19}.

Correction 2-1-3

Before correction: Main text, line 56

However, a responsivity as high as 10^6 V/W has never been realized in a bolometer operating in the radio-frequency region.

After correction: Main text, line 55

However, a responsivity as high as 10^6 V/W has never been realized in a rectification device operating in the sub-GHz frequency region as can be seen from Fig 1(a)²¹⁻²⁵.

However, one important contribution of this work is the further elucidation of the detailed mechanism of the bolometric effect. The utilization of HCMA as an important ingredient in the improved responsivity is noteworthy. However, this mechanism has already been identified and published by the authors in Ref. [27], where they extensively analyzed a very similar device in a very similar set of measurements. Again the present manuscript seems to be simply an incremental improvement over previously published work.

Response 2-2

Thank you for your comment. As you mentioned, we revealed the mechanism of HCMA in Ref(M. Goto *et al.*, Nat. Nanotechnol, **14**, 40 (2019)) using a similar experimental technique. However, since the direction of spin-torque and its dependence on magnetization direction differ between HCMA and spin-transfer torque, it is not obvious that HCMA can contribute to the enhancement of diode voltage, as we demonstrate in the present study that it does. (See also the Supplementary Information, in which we have characterize the effective rf magnetic fields of spin-transfer torque, VCMA, and HCMA and show that the rf magnetic field induced by HCMA is dominant.) This point is not simply an incremental improvement.

Moreover, further enhancement of the spin-torque induced by HCMA is possible through improving the thermal design, as discussed by Okuno (R. Okuno et al., J. Phys. Cond. Mater, 32, 384001 (2020)). On the other hand, the enhancement of spin-transfer torque requires a decrease in the magnetization or thickness of the ferromagnetic layer, which may induce deterioration of MTJs. This distinguishes our work from Ref(L. Zhang et al., Appl. Phys. Lett, (2018)) in which spin-transfer torque was used. Therefore, the use of HCMA on spin-torque diodes holds promise for the development of a highly sensitive bolometer and has also broader significance in terms of the progress of electronics.

To clarify these points, we have corrected the manuscript as follows:

Correction 2-2-1

Insertion: Main text, line 61

It was not obvious in advance that heat-induced spin-torque would provide a high responsivity exceeding that of spin-transfer torque²⁵, given the difference between the directions of their spin torques. Nevertheless, we have found this to be the case.

Correction 2-2-2

Insertion: Main text, line 189

Moreover, the HCMA value can be enhanced by improvement of thermal design as discussed by Okuno³⁰. By contrast, enhancement of spin-transfer torque requires a decrease in the magnetization or thickness of the ferromagnetic layer; this induces deterioration in MTJs. Therefore, utilization of HCMA is promising for further enhancement of responsivity.

The paper reads as a rather dry technical report rather than a Nature Communications article. The authors discuss the mechanism for the observed effects on pages 4 and 5, but many readers will find this simply to be a list of jargon that only experts in this field can appreciate or understand. Instead, I recommend that the authors introduce a new figure in the form of a creatively illustrated drawing that clearly illustrates the mechanism for the effect through un-ambiguous near-cartoonish depiction. The reader should be drawn in by

this illustration and left with a clear understanding of the basic physics that leads to the observed effect. This will also give the authors the opportunity to clearly state what is unique and original about this device design and operation, and what revolutionary new breakthrough has been made.

Response 2-3

Thank you for giving valuable comment. As you mentioned, we used a certain amount of jargon in the section “Mechanisms of the proposed spin bolometer”; we have moved it to the Supplementary Information. To replace it, we have added to the Introduction Figs. 1(b) and 1(c) and their accompanying discussion, which explain the phenomenon studied in this manuscript.

Correction 2-3-1

Insertion: Main text, Figure 1(b) and 1(c)

Figure 1 (b) (c) Schematic of spin bolometer (b) without and (c) with applied microwaves. Red and black arrows represent the magnetizations of the ferromagnetic free and pinned layers, respectively.

Correction 2-3-2

Insertion: Main text, line 68

Figures 1(b) and 1(c) show the schematics of a circuit and an MTJ without and with the application of microwaves. MTJs have the structure:

ferromagnet (pinned layer) | insulator | ferromagnet (free layer) | insulator. The magnetizations of the free and pinned layers are represented by red and black arrows. Bias voltage is applied to the MTJ, which induces the magnetization precession of the free layer due to spin-torque auto-oscillation²⁶. As shown in Fig 1(c), application of microwaves to the MTJ changes the temperature of the free layer. This induces a change in the magnetization precession, and, as a result, the MTJ's resistance is changed. With a dc bias current, this change in the resistance can be detected by the dc voltage change (see Supplementary Information for details).

I have a few more detailed concerns about the impact of these results, especially those shown in Figs. 1 and 2. First, the voltage response is strong only over a very narrow frequency range. How will the detector be made to work over a broad frequency band? Second, the voltage response is linear only over a relatively narrow dynamic range of microwave signal strength. How will the dynamic range of the detector be improved?

Response 2-4

Thank you for your comment. As shown in Fig 2(b), the diode voltage is obtained at 0.59 GHz. The full width at half maximum of 0.1 GHz increases with increasing peak frequency; however, drastic tuning is impossible. Instead, if we change the magnetic field and voltage conditions, the peak frequency can be tuned as shown in Fig R1. Consequently, microwaves can be detected at various frequencies, allowing the usable (effective) band width to be widened.

Figure R1 Frequency dependence of diode voltage under various magnetic field conditions (B , θ , φ). Red, green, and blue curves represent the diode spectra at the magnetic field conditions (50 mT, 79°, 45°), (54 mT, 78°, 50°), and (60 mT, 80°, 50°), respectively.

To enhance dynamic range, improvement of the signal to noise ratio (SNR) is necessary. In this experiment, the diode voltage reaches a noise equivalent voltage (NEV) of 0.1 mV at a microwave power of 0.1 nW. The decrease in the NEV enhances the dynamic range. The SNR can be improved by increasing ferromagnetic-layer thickness. Although spin-transfer torque and VCMA are inversely related to FeB thickness, the HCMA is more successful at keeping its magnitude even when the FeB thickness increases. The black dots in Fig R3 show how the temperature increase in an FeB layer sandwiched by MgO layers under dc bias current depends on the thickness of the layer. The temperature-increase decay curve is shallower than the $1/t_{\text{FeB}}$ function. This is because the increase of temperature of the ferromagnetic layer is mainly affected by the the MgO layer through which the heat flows, rather than the FeB layer. Therefore, the HCMA is the better torque for improving the dynamic range of responsivity.

Figure R3 Black plot represents the dependence of temperature increase in the FeB layer on the layer's thickness t_{FeB} . Dotted line is the function $1/t_{\text{FeB}}$. We assumed that the value of the function at $t_{\text{FeB}} = 3.0$ nm is the same as the simulation result

We added an explanation in the Supplementary Information as follows:

Correction 2-4-1

Insertion: Supplementary Information, line 157

Frequency tunability and dynamic range of responsivity

As shown in Fig 2(b), the diode voltage is obtained at 0.59 GHz with a line width of only 0.1 GHz. This frequency is tunable by magnetic-field and bias-voltage conditions. Figure S5 shows the diode spectra at various magnetic field conditions. The peak frequency can be modulated from 0.59 GHz to 0.74 GHz; there is a tradeoff between increasing peak frequency and decreasing responsivity.

Supplementary Figure S5 Frequency dependence of diode voltage at various magnetic field conditions (B , θ , φ). Red, green, and blue curves represent the diode spectra at the magnetic field conditions (50 mT, 79°, 45°), (54 mT, 78°, 50°), and (60 mT, 80°, 50°), respectively.

As shown in Fig. 3, the dynamic range of the diode voltage is approximately from 0.1 mV to 10 mV. To expand this dynamic range, improvement of the signal to noise ratio (SNR) is necessary. In this experiment, the diode voltage reaches a noise equivalent voltage (NEV) of 0.1 mV at the microwave power of 0.1 nW. The decrease in the NEV enhances the dynamic range. This SNR can be improved by increasing the ferromagnetic-layer thickness. Spin-transfer torque and VCMA are inversely related to the FeB thickness. HCMA also diminishes with thickness, but the decrease is smaller than that in spin transfer torque or VCMA. This is because the increase in temperature of the ferromagnetic layer is affected by the MgO layer through which the heat flows. Therefore, HCMA is preferable for improving the dynamic range of the responsivity.

Correction 2-4-2

Insertion: Main text, line 194

HCMA is also useful for enhancement of dynamic range. Although the dynamic range is limited by the noise equivalent voltage, it can be improved by increasing the ferromagnetic thickness. However, spin-transfer torque and VCMA decrease significantly when this is done. HCMA decreases only slightly because the

increase in the temperature of the ferromagnetic layer is mainly affected by the MgO layer through which the heat flows, rather than the FeB layer. Therefore, HCMA is the appropriate spin-torque for improving dynamic range (see Supplementary Information).

In addition to making changes to the manuscript made in response to the reviewers' comments, we changed the following points:

We performed English proofreading and corrected grammar and language issues.

We apologize that the horizontal axis in Fig 5(b) of previous version was not correct because we used an incorrect parameter. When this is corrected, the magnitude and sign of the bias voltage are changed, and the HCMA value is corrected to $-2.7 \mu\text{J/Wm}$. We correct these points as follows:

Correction 3-1

Before correction: Main text and Supplementary Information, HCMA value

$-2.1 \mu\text{J/Wm}$

After correction: Main text and Supplementary Information, HCMA value

$-2.7 \mu\text{J/Wm}$

Correction 3-2

Before correction: figure 5(b)

After correction: figure 6(b)

Correction 3-3

Before correction: Supplementary Information, line 14

Figure S1(b) shows the in-plane magnetic field dependence of the resistance of MTJ. The magnetic field is applied in the +y direction,

After correction: Supplementary Information, line 14

Figures S1(b) and S1(c) show the in-plane and out-of-plane magnetic field dependence of the resistance of the MTJ, respectively. The in-plane magnetic field is applied in the +y direction,

Correction 3-4

Before correction: Supplementary Information, Figure S1

Supplementary Figure S1. (b) In-plane magnetic field dependence of the MTJ resistance. The direction of magnetic field is applied in the +y direction. The solid and dashed lines represent the sweep direction of the magnetic field.

After correction: Supplementary Information, Figure S1

Supplementary Figure S1. (b) In-plane and (c) out-of-plane magnetic field dependence of the MTJ resistance. The in-plane magnetic field is applied in the +y direction. The solid and dashed lines represent the sweep direction of the magnetic field.

REVIEWERS' COMMENTS

Reviewer #1 (Remarks to the Author):

The authors have addressed the points raised by the reviewers, and the revised manuscript is prepared more clear, hence I recommend to publish this paper.

Reviewer #2 (Remarks to the Author):

The authors have largely addressed my concerns about this manuscript.

First they have clarified what makes this work distinctly different from previous work, and what level of improvement has been made. Demonstrating a high responsivity at such low frequencies is indeed a remarkable accomplishment. The HCMA mechanism is uniquely able to accomplish this goal. The authors now make this point clearly with the new Fig. 1(a). I feel more confident that this is a significant and noteworthy breakthrough.

Secondly, the explanation of the physics underlying the effect is now much more clearly stated. The jargon-filled description has been relegated to the Supp. Info. and a new figure that illustrates the basic idea now appears in Fig. 1(b) and (c). This is a big improvement.

The comments concerning improving the bandwidth and dynamic range are helpful. These are significant improvements to the text and Supp. Info.

Overall, I believe that my concerns have been addressed and I am happy to see this improved manuscript published.

REVIEWER COMMENTS

Reviewer #1 (Remarks to the Author):

The authors have addressed the points raised by the reviewers, and the revised manuscript is prepared more clear, hence I recommend to publish this paper.

Thank you for your constructive and insightful comments.

Reviewer #2 (Remarks to the Author):

The authors have largely addressed my concerns about this manuscript.

First they have clarified what makes this work distinctly different from previous work, and what level of improvement has been made. Demonstrating a high responsivity at such low frequencies is indeed a remarkable accomplishment. The HCMA mechanism is uniquely able to accomplish this goal. The authors now make this point clearly with the new Fig. 1(a). I feel more confident that this is a significant and noteworthy breakthrough.

Secondly, the explanation of the physics underlying the effect is now much more clearly stated. The jargon-filled description has been relegated to the Supp. Info. and a new figure that illustrates the basic idea now appears in Fig. 1(b) and (c). This is a big improvement.

The comments concerning improving the bandwidth and dynamic range are helpful. These are significant improvements to the text and Supp. Info.

Overall, I believe that my concerns have been addressed and I am happy to see this improved manuscript published.

Thank you for your constructive and insightful comments.

Corrections

We have complied with the formatting instructions of *Nature Communications*, and we have revised our manuscript as described herein. In addition, we have made a few revisions for readability; however, we have not changed the essential content of our manuscript.

Correction 1

We have reduced the word count of the abstract to be within 150 words.

Correction 2

We have removed the term “new.”

Before correction: Main text, Line 65

This paper describes a new and highly sensitive microwave detection technique using heat-induced spin-torque.

After correction: Main text, Line 65

This paper describes a highly sensitive microwave detection technique using heat-induced spin-torque.

Correction 3

In the explanation of film structure, we have used “/” that means “top side / bottom side” (For example, Ta/Ru). Additionally, we have also used “|” that means “bottom side | top side.” This may lead to a misunderstanding; thus, we have replaced “/” with “|”, and have rewritten the order of materials.

Main text, line 79 and 231,

(Before correction) Ta/Ru

(After correction) Ru | Ta

(Before correction) Ru(7.0 nm)/Ta(3.0 nm)

(After correction) Ta(3.0 nm) | Ru(7.0 nm)

Correction 4

We have corrected the experimental condition regarding the amplitude modulation frequency.

Before correction: Main text, Line 236

For the diode measurements, microwaves with an amplitude modulation frequency of 7.5 kHz were applied to the MTJ from a signal generator (Keysight, E8257D).

After correction: Main text, Line 236

For the diode measurements, microwaves were applied to the MTJ from a signal generator (Keysight, E8257D). The amplitude modulation frequencies of microwaves were 7.5 kHz, 1.0 kHz, and 3.0 kHz in Figs 2–3, 5(d), and 6, respectively.

Before correction: Supplementary Information, Line 31

Microwaves with an amplitude modulation frequency of 7.5 kHz were applied to the magnetic tunnel junction (MTJ) from a signal generator.

After correction: Supplementary Information, Line 31

Microwaves were applied to the magnetic tunnel junction (MTJ) from a signal generator.

Correction 5

We have changed the heading “Results and Discussions.”

Main text, line 76,

(Before correction) Results and Discussion

(After correction) Results

Correction 6

We have added subheadings in the Results section.

Main text, Line 78,

Experimental design.

Main text, Line 88,

Responsivity.

Main text, Line 109,

Noise power spectrum.

Main text, Line 136,

HCMA.

Main text, Line 201,

Noise equivalent power.

Main text, Line 230,

Sample preparation.

Main text, Line 236,

Diode measurement

Main text, Line 250,

Measurement of noise power spectrum.

Main text, Line 255,

Correspondence of samples and measurement results.

Correction 7

We have added items to the Supplementary Information (Supplementary Note 1–6).

Correction 8

We have changed the order of Supplementary Notes 1–6 such that it corresponds to the order in which they appear in the main text.

Correction 9

We have improved the resolution of Figs 1–6.

Correction 10

Other corrections regarding formatting have been written in the file “Author_checklist_toupload_1606143717_7.”